# Optimising personal continuity for older patients in general practice: a cluster randomised stepped wedge pragmatic trial

Lex Groot ![ORCID],[1] Henk Schers ![ORCID],[2] J S Burgers ![ORCID],[3] Martin Smalbrugge ![ORCID],[4] Annemarie A Uijen ![ORCID],[2] Jeroen Hoogland ![ORCID],[5] Henriëtte E van der Horst ![ORCID],[1] Otto R Maarsingh ![ORCID] [1]

[1]Department of General Practice, Amsterdam UMC Location VUmc, Amsterdam, Netherlands
[2]Department of Primary and Community Care, Radboudumc, Nijmegen, Netherlands
[3]Guideline Development and Research, Dutch College of General Practitioners, Utrecht, Netherlands
[4]Department of Medicine for Older People, Amsterdam UMC Locatie VUmc, Amsterdam, Netherlands
[5]Epidemiology and Data Science, Vrije Universiteit Amsterdam, Amsterdam, Netherlands

**Correspondence to**
Dr Otto R Maarsingh;
o.maarsingh@amsterdamumc.nl

## ABSTRACT

**Aim** To evaluate the effectiveness, feasibility and acceptability of a multicomponent intervention for improving personal continuity for older patients in general practice.

**Design** A cluster randomised three-wedged, pragmatic trial during 18 months.

**Setting** 32 general practices in the Netherlands.

**Participants** 221 general practitioners (GPs), practice assistants and other practice staff were included. Practices were instructed to include a random sample of 1050 patients aged 65 or older at baseline and 12-month follow-up.

**Intervention** The intervention took place at practice level and included opTimise persOnal cOntinuity for oLder (TOOL)-kit: a toolbox containing 34 strategies to improve personal continuity.

**Outcomes** Data were collected at baseline and at six 3-monthly follow-up measurements. Primary outcome measure was experienced continuity of care at the patient level measured by the Nijmegen Continuity Questionnaire (NCQ) with subscales for personal continuity (GP knows me and GP shows commitment) and team/cross-boundary continuity at 12-month follow-up. Secondary outcomes were measured in GPs, practice assistants and other practice staff and included work stress and satisfaction and perceived level of personal continuity. In addition, a process evaluation was undertaken among GPs, practice assistants and other practice staff to assess the acceptability and feasibility of the intervention.

**Results** No significant effect of the intervention was observed on NCQ subscales GP knows me (adjusted mean difference: 0.05 (95% CI −0.05 to 0.15), p=0.383), GP shows commitment (0.03 (95% CI −0.08 to 0.14), p=0.668) and team/cross-boundary (0.01 (95% CI −0.06 to 0.08), p=0.911). All secondary outcomes did not change significantly during follow-up. Process evaluation among GPs, practice assistants and other practice staff showed adequate acceptability of the intervention and partial implementation due to the COVID-19 pandemic and a high perceived workload.

**Conclusion** Although participants viewed TOOL-kit as a practical and accessible toolbox, it did not improve personal continuity as measured with the NCQ. The absence of an effect may be explained by the incomplete

## STRENGTHS AND LIMITATIONS OF THIS STUDY

⇒ Pragmatic trial design facilitated evaluation of the effectiveness and implementation of the intervention in a real-world setting and increased participant commitment to the study.
⇒ Robust evaluation of a complex intervention developed to promote the provision of personal continuity in general practice.
⇒ Use of a validated, patient-reported outcome measurement for personal continuity.
⇒ The study took place during the COVID-19 pandemic, which influenced study design and outcomes.

implementation of TOOL-kit into practice and the choice of general outcome measures instead of outcomes more specific for the intervention.

**Trial registration number** International Clinical Trials registry Platform (ICTRP), trial NL8132 (URL: ICTRP Search Portal (who.int)).

## INTRODUCTION

Continuity of care is a core principle of general practice.[1] It is defined by three dimensions[2]:

► Personal continuity: Having a personal provider in every separate care setting who knows and follows the patient.
► Team continuity: Exchange of relevant patient information and cooperation between care providers within one care setting to ensure that care is connected.
► Cross-boundary continuity: Exchange of relevant patient information and cooperation between care providers from different settings to ensure that care is connected.

Personal continuity is highly valued by both patients and general practitioners (GPs)[3–5] and previous studies have shown that personal continuity is associated with many benefits including higher levels of patient

and GP satisfaction,[6–9] better patient–provider relationship,[6 8] better patient quality of life,[10–12] higher medication adherence,[13–15] lower use of hospital services,[7 15–20] better health outcomes[11 13 21 22] and an overall reduction in healthcare costs.[23–25] Additionally, higher levels of personal continuity are associated with a lower mortality rate.[24–31]

Over time, changes in society and healthcare have challenged the provision of personal continuity.[32 33] There is an increasing number of older patients with chronic diseases who often receive care from various healthcare providers from different organisations putting them at higher risk for fragmentation of care.[34] At the same time, these patients may benefit more from personal continuity than younger, more healthy people.[1 15 19 35–37] Additionally, it may be more difficult for GPs to build and maintain a personal relationship with their patients as GPs increasingly work part time, organise themselves into larger group practices and are more likely to work as a locum.[38–40] Finally, the healthcare systems of some countries tend to prioritise fast and/or digital access to primary care over personal continuity.[32 41 42]

Using the UK Medical Research Council framework for complex interventions,[43 44] we developed a multicomponent intervention to opTimise persOnal cOntinuity for oLder patients in general practice, called TOOL-kit.[45] In phase 0-II, we defined the components of the intervention by performing surveys and focus groups among patients, GPs, practice assistants and practice nurses.[46] Phase III consists of the evaluation of the intervention on which we report in this study.[43 44]

The aim of this study was to evaluate the effectiveness, acceptability and feasibility of the multicomponent intervention TOOL-kit for improving personal continuity for older patients in general practice.

## METHODS
### Design and setting
We conducted a cluster randomised, three-wedged, pragmatic, controlled trial in 32 general practices in the Netherlands. We have chosen this design because we hypothesised that the intervention will do more good than harm and we do not want to withhold the intervention from participants until the end of the trial. Additionally, a stepped-wedge design improves the logistical feasibility of the trial and may contribute to successful inclusion.[45 47] During the trial, the intervention was rolled out sequentially in general practices (clusters) over three time periods (see online supplemental file 1. Data were collected at baseline (T0) and at six 3-monthly follow-up measurements (T1–T6). A detailed study protocol has been published elsewhere.[45]

The trial was registered with the International Clinical Trials Registry Platform on 2 November 2019 (trial NL8132)

We adhered to the Consolidated Standards of Reporting Trials 2010 guideline extension for stepped wedge cluster randomised trials in reporting the findings of our study.[48]

### Inclusion and participants
We included general practices (clusters) from two regions in the Netherlands: the Amsterdam and Nijmegen areas. Practices were included purposefully to ensure a varied sample with regard to level of urbanisation, practice size and socioeconomic status of the inhabitants of the practice area. Only practices with three or more regularly employed GPs were included.

From these practices, we included GPs, patients and practice assistants as participants.

At month 0, we requested practices to include a minimum of three GPs and one practice assistant at the moment of practice inclusion. All GPs and practice assistants working in permanent employment of the practice were eligible for inclusion. Other practice staff (eg, practice nurses and managers) could also participate if they were considered to be relevant stakeholders in personal continuity by a participating practice. Informed consent was acquired from GPs, practice assistants and other practice staff.

At months 0 and 12, we instructed practices to include a random sample of 100 patients. We focused on patients aged 65 or above as these may be more impacted by fragmentation of care and benefit the most from personal continuity.[1 11 35] Patient inclusion criteria were age 65 years or older, living at home, no severe cognitive disabilities, at least one consultation in the previous 12 months (including telephone calls and e-consultations) and ability to understand and write in Dutch. GPs were allowed to exclude patients for participation for whom they felt participating in the study was too high a burden. Practices were asked to manually screen the list of patients for inclusion or exclusion criteria. Samples were taken independently from each other on on months 0 and 12 .

### Intervention
The intervention consisted of TOOL-kit: a toolbox containing 34 strategies for improving personal continuity in general practice (see box 1). Contents of TOOL-kit have been detailed in online supplemental material S2. The development of TOOL-kit was guided by previous research on personal continuity in primary care. The developmental process is described in our protocol paper and online supplemental figure 1.[45] Briefly, in 2019, we conducted a survey and focus groups among GPs, patients, practice assistants and nurses to gather their views on how to improve personal continuity in older patients.[46] Additionally, we used the results of a pilot study conducted by ORM in 2017 (see online supplemental material S3) and the recommendations of the Continuity of Care Toolkit developed by the Royal College of General Practitioners in 2014 as a source of potential TOOL-kit components.[49 50] A total of 239 suggestions for improving personal continuity were identified. After deduplication and reformulation, 86 suggestions remained. These suggestions were reviewed by members of our research group (HS, JSB, MS, AAU, HEvdH and ORM) in a Delphi study. The Delphi involved two rounds

**Box 1    34 strategies to improve personal continuity in general practice from TOOL-kit***

**I. Using personal lists.**
1 Every patient is on the personal list of a GP.
2 All GPs in the practice have patients registered on their personal list.
3 Patient records in the EHR system display a pop-up message with the name of the patient's regular** GP.
**II. Instructing practice assistants.**
4 Practice assistants should preferably schedule appointments with the patient's regular GP.
5 Practice assistants should preferably schedule appointments for patients with complex problems with the patient's regular GP.
6 Follow-up telephone consultations for one illness episode are conducted by the patient's regular GP.
7 One problem, one GP.
**III. One patient, one GP.**
8 Home visits for housebound patients are conducted by one or two regular GPs.
9 Repeat prescriptions are preferably issued by the patient's regular GP.
10 Laboratory results are assessed and discussed with the patient by the requesting GP.
**IV. Working with part-time GPs.**
11 Part-time GPs work together in duos.
12 Patients with complex problems have two regular GPs.
13 Part-time GPs working in duos have regular consultation meetings.
**V. Working proactively.**
14 All GPs in the practice offer face-to-face meetings to newly registered patients aged 65 years and over.
15 All GPs in the practice have regular telephone contact with patients with complex problems.
16 The patient's regular GP contacts patients when they return home after a hospital admission.
17 Structured and regular identification of patients with low continuity of care.
**VI. Using e-health.**
18 Patients are regularly informed of organisational changes in the practice via the practice website.
19 Patients are regularly informed of organisational changes in the practice via a newsletter.
20 Patients can request e-consultations.
21 Patients can make appointments online with the healthcare provider of their choice.
**VII. Optimising accessibility and availability.**
22 GPs offer telephone consultation to their own patients on non-consultation days.
23 A call-back list for telephone consultations is managed by the practice.
24 The standard consultation time is 15 min.
25 GPs are available outside of office hours for intercollegial consultations on complex patients.
26 GPs are available for terminally ill patients outside of office hours.
27 Every regular GP offers consultation hours at least 3 days a week.
**VIII. Working with locum GPs.**
28 Minimise the number of locums.
29 GPs who share the care of a patient population do not go on holiday at the same time.
30 Locum GPs write a handover report.
**IX. Cooperating with practice healthcare workers.**
31 GPs record their medical considerations and personal reflections systematically in the EPR system.

Continued

**Box 1    Continued**

32 Practice nurses and GPs inform each other of patients' life events.
33 The patient's regular GP performs one quarterly diabetes check-up per year in their patients with diabetes by default.
34 The records of patients with complex problems contain a medical summary.

*All strategies are described more elaborately in supplementary material 2. **A 'regular GP' is defined as the same GP who is consulted most or all of the time by a patient. EHR, electronic health record; GP, general practitioner.

with 80% and 50% consensus thresholds, based on other research.[51] As a result, 34 suggestions were included in TOOL-kit and categorised into 9 domains. These suggestions were used to develop comprehensive strategies for improving personal continuity adhering to the SMART guidelines for goal-setting.[52]

The application of TOOL-kit in a practice took place in four steps:

Step 1: Run practice scan—Each practice completed a short survey with yes/no questions on current activities to promote personal continuity in the practice organisation.

Step 2: Select strategies for improvement—Based on the practice scan results, a list of recommended improvement strategies was generated by TOOL-kit. Practices could select any number of recommended strategies from this list.

Step 3: Draft a practice improvement plan—Based on the selected strategies, each practice drafted an improvement plan based on the SMART methodology. TOOL-kit includes step-by-step guidance for each strategy to support practices in writing the draft.

Step 4: Implement the practice improvement plan—Each practice implements the contents of the final improvement plan.

In addition to the 34 strategies outlined in box 1, TOOL-kit comprises three generic components for improvement of personal continuity. These components include the provision of information material to patients to educate them on the benefits of personal continuity, a digital learning module for GPs, practice assistants and other practice staff to refresh their knowledge on personal continuity and a digital manual for practices to compute and evaluate a quantitative measurement for personal continuity.[53] Practices received all necessary materials from the researchers.

During the trial, TOOL-kit was rolled out sequentially at the practice level. Wedge 1 received the intervention immediately after the baseline measurement, wedge 2 after 3 months of usual care and wedge 3 after 6 months of usual care. TOOL-kit and associated materials were sent to practices by email. Practices were invited and stimulated to complete each step of TOOL-kit. We did not provide practices with detailed instructions or criteria. Practices received only minimal technical support from researchers to facilitate self-sufficiency of TOOL-kit and simulate a real-world practice setting.

## Usual care

During the trial, usual GP care continued unrestricted in both the control and intervention condition.

## Outcome measurements

The primary outcome was a patient-reported outcome measure: patients' experienced continuity of care measured by the Nijmegen Continuity Questionnaire (NCQ).[54 55] The NCQ consists of three subscales: personal continuity: care provider knows me, personal continuity: care provider shows commitment and team/cross-boundary continuity. Scores are averaged per subscale, resulting in three continuity scores. The NCQ was distributed to patients at months 0 and 12 in all practices.

As secondary outcome measures, we asked GPs, practice assistants and other practice staff about their levels of work stress and satisfaction as changes in personal continuity may influence these outcomes.[6 9] In addition, we asked them how they perceived the level of personal continuity in their practice as a measure of face validity of intervention effectiveness. The outcomes were measured on month 0 and every 3-monthly follow-up moment by digital surveys.

## Process evaluation

A process evaluation was performed by assessing the feasibility and acceptability of the intervention at every 3-monthly follow-up measurement using the digital surveys. Feasibility was evaluated at the practice level and was determined by asking practices to estimate outcomes specific to the selected improvement strategies as follows. For each strategy displayed in box 1, a specific outcome measure was formulated that related as closely as possible to the strategy target and daily general practice. Practices were asked to provide a substantiated estimate of the specific outcome using TOOL-kit. In addition, practices were asked which parts of their practice improvement plan they had achieved and to elaborate on their answer using an open-ended question.

Acceptability was determined by longitudinally assessing GPs, practice assistants and other practice staffs' perceived priority given to improving personal continuity and their ability to do so. In addition, participants were asked '*Do you feel TOOL-kit is able to improve personal continuity?*' (yes/no) after receiving the intervention (wedge 1: T1, wedge 2: T2, wedge 3: T3) and at the 18-month follow-up measurement. Participants were asked to elaborate on their response in an open-ended question.

## Baseline characteristics

Baseline characteristics of practices and GPs, practice assistants and other practice staff were collected. Practice characteristics included the number of employees, the average full-time equivalent and level of urbanisation. Practice employee characteristics included age, sex, employment status, working experience and time in clinical activities per week. Patient characteristics were collected together with the NCQ during each cycle and included age, sex, the number of GP practice visits in the past 12 months and the number of years registered with the practice.

## Randomisation and blinding

Practices were randomised to a wedge with computer-generated random numbers using Microsoft Excel 2016. Randomisation was performed by one of the researchers (LG). Due to the study design and nature of the intervention, researchers, participants and patients could not be blinded to wedge assignment during the trial. All researchers were blinded to wedge assignment during data analysis.

## Sample size

Sample size calculation was based on the primary outcome measure (NCQ). For both NCQ measurements at months 0 and 12, we expected at least 35 patients per practice and 30 practices would participate, resulting in a total of 1050 patients for each measurement. Using a mixed linear model with a random effect for practice and two-sided testing at a significance level of 5%, the expected sample size allowed detection of all standardised effect sizes of 0.12 and larger with at least 80% power assuming an intracluster correlation coefficient (ICC) of 0.01. The standardised effect size was defined as the difference in means divided by the total SD incorporating both the between-practice and within-practice variation.

## Statistical analysis

For the primary outcome (continuous), we used linear mixed model analysis to compare the intervention condition (TOOL-kit and usual care) to the control condition (usual care only). We established a crude model with a random intercept at the practice level to account for clustered inclusion of patients. No intercept on the patient level was added as no repeated measures took place on the individual level. Baseline measurements were used in the random effects model to adjust for differences in baseline due to clustering. The final model was adjusted for clustering and differences in exposure to TOOL-kit between wedges. No further adjustments were made. Heterogeneity of intervention effect due to longer exposure to TOOL-kit between wedges and temporal effects were investigated using a two-way interaction term.

For patient data, there were missing data on items of the NCQ. We used multiple imputation to handle missing data before calculation the NCQ subscales using all variables in the analysis models.[56] These variables were chosen to make optimal use of the observed subscale data. In total, 5 imputed sets were derived using fully conditional specification with 50 iterations. Results were pooled using Rubin's rules.[57]

For the secondary outcomes (ordinal), we used a cumulative link mixed model for outcomes on a 5-point or 10-point scale to compare the intervention condition to the control condition. In the crude model, adjustments were made for clustering of observations on both

the practice and individual level by means of random intercepts. The final model was adjusted for clustering, repeated measurements between individuals, baseline differences, differences in exposure to TOOL-kit between wedges, calendar time and baseline differences between wedges. Additional procedures for missing data were not required for this analysis.[58] For the primary and secondary outcome measures, crude and adjusted mean differences and ORs with their 95% CIs are presented. All statistical analyses were performed by using SPSS (V.28). All tests and reported p values are two sided with α=0.05.

Process evaluation outcomes were analysed with descriptive statistics (mean, percentage, count). Here, 5-point and 10-point scale items were trichotomised for purposes of analysis. Answers to open-ended questions were analysed qualitatively. Here, LG subsequently read, grouped and categorised the responses and identified overlying concepts.

### Patient and public involvement
GPs, practice assistants and practices nurses and patients were involved in the conception and the development of TOOLkit. Before the study started, we performed a pilot study among 20 GPs in Amsterdam, asking for input on the study topic. In phase I, we performed a survey and focus group among patients to collect their views on improving personal continuity.[45] HS, JSB, AAU, ORM and HEvdH are GPs and were involved in the development of the research question, study design and choice of outcome measures as public contributors.

No patients were involved in the development of the research question, the study design or the choice of outcome measures. During the conception of TOOLkit, it became apparent that the intervention would not target patients but practice organisation. As patients have less insight into general practice organisation, a limited involvement of patients in this project was more appropriate given the contents of the intervention.

### RESULTS
The flow of practices and participants through the study is displayed in figure 1. Practices were recruited between October 2019 and March 2020. A total of 221 GPs, practice assistants and other practice staff from 32 general practices were included (median practice staff per practice: 6, IQR 4–8). During the trial, three practices withdrew: one before baseline measurement and two during follow-up. All withdrawing practices reported an increased workload due to the COVID-19 pandemic and/or practice reorganisation as reasons for withdrawal. A total of 39 GPs, practice assistants or other staff were lost to follow-up due to the participant leaving the practice (n=16), withdrawal of the practice itself (n=13), chronic illness (n=5), retirement (n=4) or loss of motivation for study participation (n=1).

As 1 of the 32 practices withdrew before month 0 (baseline), we included 3100 patients at month 0 (baseline). At month 12 (follow-up), 3 of the 32 practices had withdrawn and we included 2900 patients. NCQ response rates were 39% (1210/3100) at baseline and 38% (1109/2900) at follow-up.

Characteristics of practices, participants and patients are displayed in table 1 Characteristics were collected from 210/221 GPs, practice assistants or other staff (response rate: 95%).

### Primary outcome: patients experienced continuity of care
The results for the primary outcome measure are shown in table 2. Descriptive statistics shows small baseline differences ('GP knows me' mean: 0.02 (SD: 0.17), 'GP shows commitment': 0.05 (0.25), team/cross boundary: 0.03 (0.13)) and small differences between baseline and 12-month follow-up measurements on all three NCQ subscales. After adjusting for baseline differences and exposure to TOOL-kit, no significant effect of the intervention on perceived personal continuity was observed for the subscales GP knows me (adjusted mean difference: 0.05 (95% CI −0.05 to 0.15), p=0.383) and GP shows commitment (0.03 (95% CI −0.08 to 0.14), p=0.668). The team/cross-boundary continuity subscale also showed no relevant differences (0.01 (95% CI −0.06 to 0.08), p=0.911). The ICCs for the subscales were 0.07, 0.05 and 0.04, respectively. Practice area and the number of GPs, assistants or nurses per practice were examined as possible confounders. While they did slightly modify the point estimate for the treatment effect (online supplemental table S1), these changes did not alter the main conclusion.

### Secondary outcomes: healthcare worker stress, work satisfaction and perceived personal continuity
Among GPs, practice assistants and other practice staff, the mean overall response rate to the digital surveys was 80%, ranging from 95% at T0 (210/221) to 67% at T6 (121/182). The results for the secondary outcomes are displayed in table 3. An increase in work stress level was observed in the crude analysis of the intervention effect (crude OR 2.28 (95% CI 1.70 to 3.07). After adjusting for differences in exposure to TOOL-kit and time, the effect dissipated (adjusted OR 1.55 (95% CI 0.80 to 2.98). Adjusted analyses showed no intervention effect on work satisfaction or perceived personal continuity in a practice.

### Process evaluation
As one practice from wedge 3 withdrew before the intervention started, 31/32 practices completed all four parts of the intervention. The results of the strategy implementation are presented in online supplemental table S2. A mean of 5.3 strategies (range: 1–9 strategies) were selected by the practices. At 18-month follow-up, 78% (128/164) of all selected strategies were still being used. Reasons to discontinue strategies were practice withdrawal from the trial (n=12), loss of relevance to practice organisation (n=8), no time/other priorities (n=7), or no reason specified (n=9).

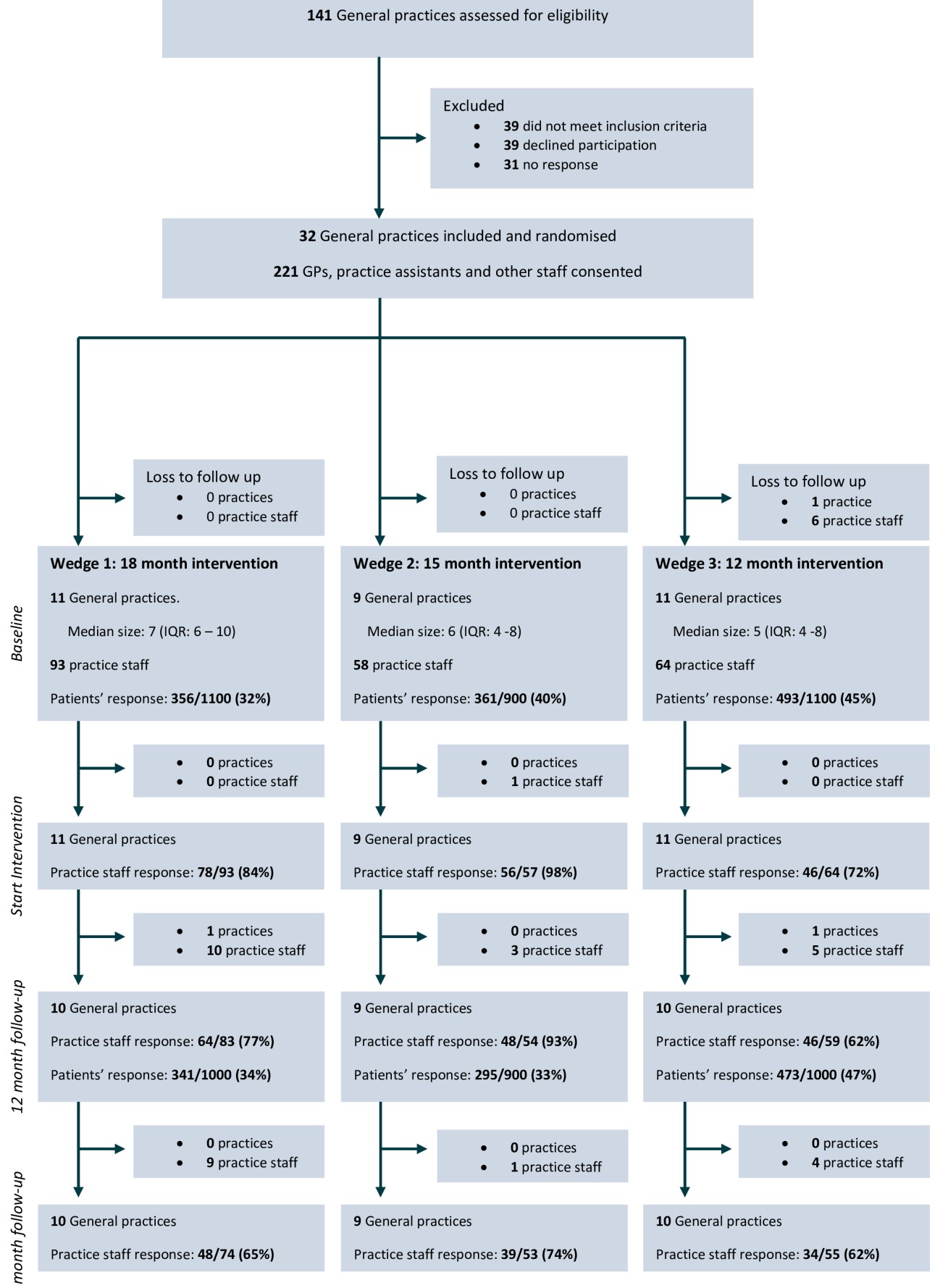

**Figure 1** Flow of practices and participants through the study. GPs, general practitioners.

**Table 1** Characteristics of general practices and participants

| General practices | | Participants | | | | |
| --- | --- | --- | --- | --- | --- | --- |
| | | GPs, practice assistants and others* | | Patients | | |
| | Baseline (n=32) | | Baseline (n=210) | | Baseline (n=1210) | Follow-up (n=1109) |
| | n (%) | | n (%) | | n (%) | n (%) |
| Number of GPs (salaried or partner) | | Sex | | Sex | | |
| 1–3 | 8 (25) | Male | 51 (24) | Male | 597 (49) | 543 (49) |
| 4–6 | 16 (50) | Female | 158 (75) | Female | 609 (50) | 565 (50) |
| 7–8 | 8 (25) | Other | 1 (1) | Other | 4 (1) | 1 (1) |
| Number of practice assistants | | Age | | Age | | |
| 1–3 | 7 (22) | 20–39 | 51 (29) | 65–69 | 389 (32) | 330 (30) |
| 4–6 | 13 (41) | 40–49 | 48 (23) | 70–79 | 618 (51) | 565 (51) |
| 7–9 | 10 (31) | 50–65 | 53 (25) | ≥80 | 203 (17) | 214 (19) |
| 10 or more | 2 (6) | Did not disclose | 48 (23) | | | |
| Number of practice nurses | | Job role | | Years registered with practice | | |
| 1–3 | 16 (50) | GP | 129 (61) | <1 | 14 (1) | 17 (1) |
| 4–6 | 13 (41) | Practice assistant | 63 (30) | 1–4 | 104 (9) | 84 (8) |
| 7–9 | 2 (6) | Practice nurse | 12 (6) | 5–10 | 135 (11) | 117 (11) |
| 10 or more | 1 (3) | Other* | 6 (3) | >10 | 957 (79) | 891 (80) |
| Mean FTE of all employees | | Years at current practice | | Number of GP consultations in the past 12 months | | |
| <0.4 | 0 (0) | <5 | 76 (36) | 1–2 | 430 (35) | 376 (34) |
| 0.4–0.6 | 21 (65) | 5–10 | 58 (28) | 3–5 | 432 (36) | 374 (34) |
| 0.7–0.9 | 4 (13) | 11–20 | 46 (22) | >5 | 348 (29) | 359 (32) |
| ≥1.0 | 7 (22) | 21–30 | 24 (11) | | | |
| | | >30 | 6 (3) | | | |
| Characteristics of practice area | | Direct patient care per week (FTE) | | | | |
| (Very) urban | 19 (59) | 0.1–0.4 | 65 (31) | | | |
| Moderate | 8 (25) | 0.5–0.8 | 136 (65) | | | |
| (Very) rural | 5 (16) | 0.9–1.0 | 9 (4) | | | |

*Practice manager (n=5), physician assistant (n=1).
FTE, full-time equivalent; GP, general practitioner.

The feasibility outcomes are detailed in online supplemental table S3. For 9 out of 34 strategies, a mean increase in the estimated specific outcome was reported together with full or partial completion of the practice improvement plan in most of the participating practices. For 13/34 strategies, there was no change or a decrease in the mean outcome measure and most practices had only partially completed their improvement plan. No parts of a practice improvement plan were completed for 9/34 strategies and practices reported a decreased or unchanged estimated specific outcome for these strategies. The remaining three strategies were not applied by any practice.

Acceptability outcomes are displayed in online supplemental table S4. Most GPs, practice assistants and other practice staff felt they had moderate to (very) high ability

to improve personal continuity in their practice both at baseline (96%) and at 18-month follow-up (95%). At the start of the trial, 99% of participants gave a moderate to (very) high priority to improving personal continuity. At 18-month follow-up, this had decreased to 89%.

Immediately after receiving TOOL-kit (n=163) and at 18-month follow-up (n=122), 45%–47% of participants agreed with the statement that TOOL-kit would be able to improve personal continuity, 2% disagreed and 52%–51% did not know. In the open-ended question, participants indicated that they perceived TOOL-kit as a flexible, practical and accessible tool. They felt TOOL-kit may increase awareness of the importance of personal continuity among healthcare workers, improve cooperation within the practice, increase patient satisfaction and benefit the overall quality of care. Participants mentioned

**Table 2** Effect of the intervention on continuity of care as measured with the Nijmegen Continuity Questionnaire

| Outcome | | | Descriptives | | Mixed model analysis | |
|---|---|---|---|---|---|---|
| | | | Mean (SD) | | Crude mean difference* (95% CI) | Adjusted mean difference† (95% CI) |
| | | | Baseline n=1210 | Follow-up n=1109 | Follow-up versus baseline | |
| Personal continuity: GP knows me | Pooled | | 3.71 (0.78) | 3.73 (0.77) | 0.02 (−0.04 to 0.09) | 0.05 (−0.05 to 0.15) |
| | Wedge | 1 | 3.67 (0.81) | 3.69 (0.77) | | |
| | | 2 | 3.82 (0.71) | 3.82 (0.77) | | |
| | | 3 | 3.64 (0.80) | 3.71 (0.76) | | |
| Personal continuity: GP shows commitment | Pooled | | 3.27 (0.85) | 3.30 (0.83) | 0.03 (−0.04 to 0.10) | 0.03 (−0.08 to 0.14) |
| | Wedge | 1 | 3.23 (0.85) | 3.24 (0.82) | | |
| | | 2 | 3.44 (0.80) | 3.48 (0.80) | | |
| | | 3 | 3.15 (0.88) | 3.25 (0.83) | | |
| Team/cross-boundary continuity | Pooled | | 3.48 (0.51) | 3.49 (0.51) | 0.01 (−0.03 to 0.06) | 0.01 (−0.06 to 0.08) |
| | Wedge | 1 | 3.46 (0.50) | 3.45 (0.52) | | |
| | | 2 | 3.57 (0.56) | 3.56 (0.56) | | |
| | | 3 | 3.41 (0.52) | 3.47 (0.47) | | |

All subscales are measured on a 5-point Likert scale (1: very low. 5: very high).
*Adjusted for cluster.
†Adjusted for cluster. exposure to TOOL-kit, baseline differences.
GP, general practitioner.

**Table 3** Effect of the intervention on the secondary outcome measurements

| Outcome | | | Mean (SD) | | | Mixed model analysis | |
|---|---|---|---|---|---|---|---|
| | | | Baseline n=161 | Start intervention n=180 | 18-month Follow-up n=121 | Crude OR* (95% CI) | Adjusted OR† (95% CI) |
| Stress‡ | Pooled | | 2.92 (0.65) | 3.18 (0.62) | 3.36 (0.67) | 2.28 (1.70 to 3.07) | 1.55 (0.80 to 2.98) |
| | Wedge | 1 | 2.88 (0.61) | 3.03 (0.56) | 3.27 (0.57) | | |
| | | 2 | 2.82 (0.52) | 3.27 (0.61) | 3.64 (0.78) | | |
| | | 3 | 3.15 (0.86) | 3.24 (0.69) | 3.15 (0.56) | | |
| Work satisfaction‡ | Pooled | | 3.22 (0.46) | 3.27 (0.49) | 3.22 (0.519) | 0.86 (0.58 to 1.27) | 1.26 (0.53 to 2.98) |
| | Wedge | 1 | 3.22 (0.50) | 3.34 (0.51) | 3.30 (0.51) | | |
| | | 2 | 3.22 (0.42) | 3.23 (0.52) | 3.20 (0.51) | | |
| | | 3 | 3.24 (0.43) | 3.24 (0.43) | 3.18 (0.55) | | |
| Perceived personal continuity§ | Pooled | | 7.15 (1.12) | 7.03 (0.95) | 6.99 (1.39) | 0.89 (0.68 to 1.16) | 1.13 (0.64 to 2.03) |
| | Wedge | 1 | 7.00 (1.18 | 7.21 (0.77) | 7.00 (1.57) | | |
| | | 2 | 7.42 (0.95) | 7.06 (0.98) | 7.05 (1.28) | | |
| | | 3 | 7.09 (1.16) | 6.81 (1.11) | 6.91 (1.26) | | |

*Adjusted for cluster, repeated measurements within individuals.
†Adjusted for cluster, repeated measurements within individuals, exposure to TOOL-kit, calendar time and baseline differences.
‡Measured on a 5-point Likert scale: 1=very low, 5=very high.
§measured on a 10-point scale: (1= very low. 10= very high).
GP, general practitioner.

the COVID-19 pandemic, high workload and more time required than anticipated as reasons why TOOL-kit was not able to improve personal continuity in their practice.

## DISCUSSION

We performed a cluster randomised pragmatic trial to investigate the effectiveness, acceptability and feasibility of an intervention to improve personal continuity for older patients in general practice (TOOL-kit). We found that TOOL-kit did not improve patients' experienced continuity nor did it affect work stress, work satisfaction or perceived level of personal continuity among GPs, practice assistants and other practice staff. As shown by the process evaluation, GPs, practice assistants and other practice staff viewed TOOL-kit as an acceptable instrument fitting the context of everyday general practice. Yet implementation of TOOL-kit in practice was often perceived as infeasible because of a high perceived workload, exacerbated by the COVID-19 pandemic.

A main strength of our study is its low number of loss to follow-up. Only 3/32 practices withdrew from the trial despite a perpetually high GP workload,[59] less engagement in scientific research among GPs[60] and an ongoing COVID-19 pandemic.[61] This may indicate a high level of commitment of practices to the subject of research. In addition, robust assessment of the intervention was possible due to a large number of patient responses and frequent measurements among GPs, practice assistants and other practice staff resulting in a large amount of data on outcome measures, acceptability and feasibility of the intervention. Finally, the pragmatic trial design facilitated evaluation of the effectiveness and implementation of the intervention in a real-world setting.[62]

A limitation of our study was the timing and frequency of the NCQ measurements. In stepped-wedge designs, measurements in control conditions are performed earlier than intervention measurements and, as such, calendar time can be a confounder. Initially, we planned to undertake two NCQ follow-up measurements instead of one. However, due to the COVID-19 pandemic and its implications for daily general practice, this number of follow-up measurements was regarded as too high of a burden for the participants by both the participating practices and the researchers. Consequently, in the analysis, we could not separate the effect of time from the effect of the intervention on personal continuity. However, given the absence of an effect, the attribution question is less relevant, and therefore, does not influence the overall conclusion of our study.

It is well known that implementation takes time.[63] Therefore, the follow-up time in our study may have been too short for adequate evaluation of the intervention. This may be especially true for wedges 2 and 3 in which practices only had 15 or 12 months of intervention time, respectively, compared with 18 months for wedge 1. However, we found no correlation between intervention time and any outcome measure. In addition, wedge 1 practices did not perform any better on any of the outcome measures than the other wedges. Therefore, it is unlikely that the limited follow-up duration of the last two wedges significantly impacted results.

In our study, there was a discrepancy between the level of the intervention (general practice organisation) and the level of the primary outcome measurement (patient). Our choice for a generic patient-reported outcome measure had to be made before the content of the intervention was defined, as is often the case with complex interventions.[44] However, changes made on practice organisational level have little influence on patients' perception of personal continuity.[46 64] In addition, using generic outcomes as opposed to outcomes specific to an intervention may limit the ability to detect changes resulting from a complex intervention.[44] This is supported by the results of our process evaluation using intervention-specific outcomes, in which we found that some strategies were effective as opposed to an overall absence of effect found with the NCQ. This partial success of the intervention-specific outcomes suggests that TOOL-kit does have the potential to orient practice organisation more towards providing personal continuity. Therefore, the results of measuring patients' experience of continuity of care with the NCQ may reflect an ecological discrepancy and not reflect the complete effectiveness of a complex intervention on the practice organisational level, such as TOOL-kit.

As the found effect sizes in our study are lower than the effect size in our power calculation, it is possible that the absence of statistical significant findings is a false negative. However, clinical significance should outweigh statistical significance. There is no set minimally clinical significant difference for the NCQ but the very small effect sizes ($\leq 0.05$ on a 5-point scale) are unlikely to constitute a clinically significant effect.

The open-ended questions were analysed only by LG. The validity of these findings could have been improved by adding a second evaluator.

Our study is the first to evaluate a complex intervention designed to improve personal continuity in general practice. Earlier studies have reported on the effectiveness of interventions for improving continuity of care for improving personal continuity. In an Australian randomised controlled trial, the effect of a patient enrolment and financial incentive did not improve personal continuity.[65] Another Australian study investigating a multicomponent intervention for improving health outcomes showed that personal continuity as measured by the usual provider continuity index actually decreased by 0.9 percentage points in their intervention group, as opposed to a 3 percentage point decrease in the control group (OR 1,14, 95% CI 1.01 to 1.28).[66] A German programme for strengthening primary care was associated with higher personal continuity scores in an observational study.[67]

We found that usage of TOOL-kit did not result in an improvement in patients' experienced personal

continuity. As mentioned above, this may be the result of our choice to use a generic outcome measure, as opposed to outcomes specific for the intervention components, and incomplete implementation which is often the reason why complex interventions fail to demonstrate effect.[68] In addition, the baseline performance of practices on the chosen strategies was already high, which may have limited the extent in which TOOL-kit could improve practice organisation. According to GPs, practice assistants and other practice staff, TOOL-kit was found to be highly acceptable and to have potential for optimising the provision of personal continuity in general practice organisations which may justify implementation of TOOL-kit into general practice.

We have previously shown that practice size, staff shortage, high turnover among staff and health insurance policy may negatively influence the provision of personal continuity for older patients in general practice.[45] However, the 34 strategies in TOOL-kit have limited influence on these factors. It is plausible that sufficient staff and adequate support from health insurance companies are requirements for providing personal continuity and effective use of TOOL-kit. The mixed results of the strategy-specific outcomes may also be in part explained by the variance of these means varies among practices. Therefore, future policy aimed at promoting personal continuity should focus on a breadth of factors on the level of the individual GP, practice organisation and health regulations.

As our study took place during the COVID-19 pandemic, daily practice changed significantly during the study period. Many practices experienced a higher workload during the pandemic, which can result in a lower commitment of a practice to the intervention.[59] However, the pandemic also facilitated the implementation of e-health by removing previously experienced barriers by GPs.[69] It was not possible for us to foresee or completely mitigate the impact of the pandemic on our study results, which may have led to changes in our assessment of personal continuity.

In future research, we, therefore, plan to expand on the process evaluation and examine potential barriers and facilitators for improving personal continuity and implementing the TOOL-kit in general practice. This expanded process evaluation will also investigate the effect of the COVID-19 pandemic on the implementation of TOOL-kit more in-depth.

**Contributors** HS, JSB, MS, AAU, HEvdH and ORM obtained funding for the trial and LG, HS, JSB, MS, AAU, HEvdH and ORM contributed to the design of the trial. LG, ORM and HS were involved in data collection and study coordination. JH and LG performed the statistical analyses. LG produced the first draft of the manuscript, all authors contributed to reviewing and editing this manuscript and approved the final version. ORM is the guarantor. The corresponding author attests that all listed authors meet authorship criteria and that no others meeting the criteria have been omitted.

**Funding** This work was supported by the Netherlands Organisation for Health Research and Development (ZonMw, programme General Practice and Old Age Medicine, registration number: 839110023).

**Competing interests** All authors have completed the ICMJE uniform disclosure form at www.icmje.org/coi_disclosure.pdf and declare no support from any organisation for the submitted work. HEvdH is a paid chair of the supervisory board of the Dutch College of General Practitioners. ORM has received various ZonMW grants for other research projects unrelated to the current study.

**Patient and public involvement** Patients and/or the public were not involved in the design, or conduct, or reporting, or dissemination plans of this research.

**Patient consent for publication** Not applicable.

**Ethics approval** This study involves human participants but Medical Ethical Committee Amsterdam UMC, VU medical centre. number: 2019-3659 exempted this study. Participants gave informed consent to participate in the study before taking part.

**Provenance and peer review** Not commissioned; externally peer reviewed.

**Data availability statement** Data are available on reasonable request. Requests for access to data from the study should be sent to the corresponding author (o.maarsingh@amsterdamumc.nl). The study protocol has been published elsewhere (Groot et al., *BMC Fam Pract*, 2021;22:207). All proposals requesting data access will need to specify how the data will be used and will need approval of the project coinvestigator team before data release.

**ORCID iDs**
Lex Groot http://orcid.org/0000-0003-1129-3315
Henk Schers http://orcid.org/0000-0002-9362-9451
J S Burgers http://orcid.org/0000-0002-6040-4221
Martin Smalbrugge http://orcid.org/0000-0003-0538-4843
Annemarie A Uijen http://orcid.org/0000-0002-7703-6250
Jeroen Hoogland http://orcid.org/0000-0002-2397-6052
Henriëtte E van der Horst http://orcid.org/0000-0003-4060-4354
Otto R Maarsingh http://orcid.org/0000-0002-3747-9217

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
