## [Reviewer comments · BMJ Open]

ARTICLE DETAILS

TITLE (PROVISIONAL)	Optimising personal continuity for older patients in general practice: a cluster randomised stepped wedge pragmatic trial
AUTHORS	Groot, Lex; Schers, Henk; Burgers, JS; Smalbrugge, Martin; Uijen, Annemarie; Hoogland, Jeroen; van der Horst, Henriëtte; Maarsingh, Otto

VERSION 1 – REVIEW

REVIEWER	Dawes, Martin University of British Columbia, Family Practice
REVIEW RETURNED	25-Sep-2023

GENERAL COMMENTS	I appreciate the in-depth response to all the reviewer's comments. I maintain my positive stance about this. It is a very useful example of the step wedge design in primary care and has many aspects that would be useful to other researchers contemplating primary care interventional research. There is much evidence from observational studies that patients receiving high levels of continuity of care, in whichever domains are examined, frequently have better outcomes. The abstract might be improved by having one or two references to this data. I also worry that by measuring continuity the researchers missed the bigger opportunity of evaluating patient outcome data. I also delved slightly deeper into the metrics and seeing that 80% of patients were getting their repeat prescriptions from the same physician was very high. Given the very high rates of continuity at zero I am not so surprised by the results. These comments do not dampen my enthusiasm for seeing this paper being published as it stands, or with minor editorial changes. It shows so clearly that challenging interventional research in primary care can be done. It also demonstrates the need for this research. This negative result illustrates how easy it is to imagine an intervention in primary care would lead to change when that is not necessarily the case.
---

REVIEWER	Campbell, John University of Exeter, Primary Care
REVIEW RETURNED	01-Oct-2023

GENERAL COMMENTS	Thank you for the opportunity to see and review the revised manuscript and the three reviews; the authors have undertaken a comprehensive and careful response and I am satisfied that all key points have been adequately addressed.
---

REVIEWER	Zeng, Irene Auckland University of Technology Faculty of Health and Environmental Sciences, Health Faculty research office
REVIEW RETURNED	19-Oct-2023

GENERAL COMMENTS	It is a very interesting study using a design that allows pragmatic method to evaluate a unit level program. The statistical analysis needs to consider methods for the complexed SWR-cluster design, with methodological references. Review summary 1. It will be in expectation to add brief justification t of using SWR-cluster design, i.e., how to mitigate known bias from SWR-cluster, referencing the protocol. (https://www.bmj.com/content/350/bmj.h391) 2. Elements of SWR-cluster design need to be defined:  -Clusters and participant -Randomisation and masking -Procedures -Outcomes 3. Statistical analysis: effect of time effect adjustment is not presented in the method.  -Apart from baseline measures used in the adjustment, any other variables used? temporal effects/time effect (i.e., calendar time) should be included in the adjustment. -Multiple imputation requires more details, what variables used for imputing the NCQ subscales and why? Please include a reference for the imputation method (i.e., Rubin). -Mixed models have not specified distribution of outcome differently for continuous outcome and ordinal outcome. For Example, Linear mixed model for continuous outcome, Generalized linear mixed or generalized estimating equations (GEE) for ordinal outcome. -If there are same participants included in more than 1 wedge period, random effect should consider include individual participant. The mixed-effect models are highly sensitive to departures from the model assumption, including the random effects. Solutions could be including a time-period random effect. https://www.ncbi.nlm.nih.gov/pmc/articles/PMC5600088/ 4. Results: Table 2. and 3 should include temporal effects/time effect (i.e., calendar time) in NCQ subscales, to derive the adjusted mean and odds ratio.
--

VERSION 1 – AUTHOR RESPONSE

Reviewer 1: Dr. Martin Dawes

There is much evidence from observational studies that patients receiving high levels of continuity of care, in whichever domains are examined, frequently have better outcomes. The abstract might be improved by having one or two references to this data.

The reviewer suggested adding references to the abstract. We think this might be a misunderstanding, as it is not custom to add references to an abstract. Reviewing recent *BMJ open* publications, we observe that *BMJ open* does not make an exception to this practice.

To process the reviewers suggestion we have adjusted the *introduction* of the manuscript: (p3, line 93-94):

“Personal continuity is highly valued by both patients and general practitioners (GPs) 3-5 and previous studies have shown that personal continuity is associated with many benefits including: [...], better health outcomes (Svereus, 2017; Cabana, 2004; Chen, 2017; Hussey, 2014), and an overall reduction in healthcare costs.”

I also worry that by measuring continuity the researchers missed the bigger opportunity of evaluating patient outcome data.

We agree with the reviewer that a more broad evaluation of patient outcome data, beside personal continuity on the patient level with the NCQ, may have contributed to a more complete overview of intervention effects. However, before starting our intervention we deliberately choose to not measure other patient related outcomes because TOOL-kit primarily targets practice organisation. Therefore, we did not find it plausible that intervention penetration in 18 months would be sufficient to provide a relevant contribution to patient outcomes.

I also delved slightly deeper into the metrics and seeing that 80% of patients were getting their repeat prescriptions from the same physician was very high. Given the very high rates of continuity at zero I am not so surprised by the results.

We agree with the reviewer that it is intriguing to see that many components of TOOL-kit had already been (partially) implemented at study start. It raises the question to what extent TOOL-kit was of added value for these practices and what dosage of the intervention was delivered. We have mentioned this in the discussion section (p20, lines 424-432). We intend to further investigate these observations in a more elaborate process evaluation.

These comments do not dampen my enthusiasm for seeing this paper being published as it stands, or with minor editorial changes. It shows so clearly that challenging interventional research in primary care can be done. It also demonstrates the need for this research. This negative result illustrates how easy it is to imagine an intervention in primary care would lead to change when that is not necessarily the case.

We highly appreciate the reviewer's positive remarks regarding the content and relevance of our paper and we agree with the demonstrated need to conduct this type of research.

Reviewer 2: Prof. John Campbell

Thank you for the opportunity to see and review the revised manuscript and the three reviews; the authors have undertaken a comprehensive and careful response and I am satisfied that all key points have been adequately addressed

We thank the reviewer for his positive remarks and are pleased that all his concerns have been addressed.

Reviewer: 3 Dr. Irene Zeng

It will be in expectation to add brief justification of using SWR-cluster design, i.e., how to mitigate known bias from SWR-cluster, referencing the protocol.

We have adjusted the manuscript to include a justification for the used study design, with reference to the study protocol (p4, lines 122-125):

“We have chosen for this design because we hypothesised that the intervention will do more good than harm and we do not want to withhold the intervention from participants until the end of the trial. Additionally, a stepped-wedge design improves the logistical feasibility of the trial and may contribute to successful inclusion. (Groot et al., 2021; Brown, 2016)”

2. Elements of SWR-cluster design need to be defined: Clusters and participant, Randomisation and masking, Procedures, Outcomes

In the light of the reviewer’s remarks, we adjusted the manuscript to more clearly define the requested elements based on the CONSORT 2010 guideline extension for stepped wedge cluster randomised trials. We have attached an updated guidance as a supplementary file to our manuscript (#10 CONSORT extension SWCR checklist_complete.pdf)

Clusters are defined as practices (p5, line 134). Participants are defined as patients, GPs and practice assistants of these practices (p5, line 138).

Randomisation took place on the level of the clusters (p10, lines 236-240). We did not perform any masking. The intervention was rolled out sequentially among practices in a stepped-wedge fashion. The primary outcome was measured on patient level and there were no repeated measurements (p9, lines 200-205). Secondary outcomes were measured on the level of practice staff and were measured repeatedly on the individual level (p9, lines 207-211).

3. Statistical analysis: effect of time effect adjustment is not presented in the method.

A. Apart from baseline measures used in the adjustment, any other variables used?

We thank the reviewer for the suggestion to further clarify the used variables for adjustment. The primary outcome was only adjusted for clustering and baseline differences. We have adjusted the manuscript to clarify this. (p11, 257-258):

“The final model was adjusted for clustering and baseline differences and differences in exposure to TOOL-kit between wedges. No further adjustments were made.”

B. Temporal effects/time effect (i.e., calendar time) should be included in the adjustment.

We agree with the reviewer that the possible bias due to temporal effects is an important consideration in stepped wedge designs. In line, we have adjusted for temporal effects where possible as is best practice. However, due to the COVID-19 pandemic and its implications for daily general practice it was not possible to perform a sufficient amount of follow-up measurements for the

primary outcome to be able to correct for calendar time. We were aware of this limitation and had mentioned this in the discussion (p18-19, lines 381-389):

“A limitation of our study was the timing and frequency of the NCQ measurements. In stepped-wedge designs, measurements in control conditions are performed earlier than intervention measurements and, as such, calendar time can be a confounder. Initially, we planned to undertake two NCQ follow-up measurements instead of one. However, due to the COVID-19 pandemic and its implications for daily general practice this number of follow-up measurements was regarded as too high of a burden for the participants by both the participating practices and the researchers. Consequently, in the analysis we could not separate the effect of time from the effect of the intervention on personal continuity. However, given the absence of an effect, the attribution question is less relevant and therefore does not influence the overall conclusion of our study.”

For the secondary outcomes, we were able to adjust the analysis for calendar time. We have adjusted the manuscript to make this more clear (p11, lines 268-270):

“The final model was adjusted for clustering, repeated measurements between individuals, baseline differences, differences in exposure to TOOL-kit between wedges, calendar time and baseline differences between wedges.”

Multiple imputation requires more details, what variables used for imputing the NCQ subscales and why? Please include a reference for the imputation method (i.e., Rubin).

We have adjusted the manuscript to make it more clear which variables were used, why and referenced Rubin's Rules (p11, lines 261-264):

“We used multiple imputation to handle missing data before calculation the NCQ subscales using all variables in the analysis models. (Heijmans, 2018) These variables were chosen to make optimal use of the observed subscale data. In total, 5 imputed sets were derived. Results were pooled using Rubin's rules. (Rubin, 1987)”

Mixed models have not specified distribution of outcome differently for continuous outcome and ordinal outcome. For Example, Linear mixed model for continuous outcome, Generalized linear mixed or generalized estimating equations (GEE) for ordinal outcome.

We have adjusted the manuscript to make this clearer (p11, line 253-254):

“For the primary outcome (continuous), we used linear mixed model analysis to compare the intervention condition (TOOL-kit and usual care) to the control condition (usual care only).”

And (p11, lines 265-266):

“For the secondary outcomes (ordinal), we used a cumulative link mixed model for outcomes on a 5- or 10-point scale to compare the intervention condition to the control condition.”

If there are same participants included in more than 1 wedge period, random effect should consider include individual participant. The mixed-effect models are highly sensitive to departures from the model assumption, including the random effects. Solutions could be including a time-period random effect. <https://www.ncbi.nlm.nih.gov/pmc/articles/PMC5600088/>

We thank the reviewer for her critical remark. For the secondary outcome, we have corrected for the repeated measures by adding a random intercept on the individual level (p11, line 266-268). This is in line with the recommendation on page five from the article referenced by the reviewer.

For the primary outcome, this procedure is not applicable as no repeated measures took place. We have adjusted the manuscript to make this clear (p11, lines 256-257):

“No intercept on the patient level was added as no repeated measures took place on the individual level.”

Results: Table 2. and 3 should include temporal effects/time effect (i.e., calendar time) in NCQ subscales, to derive the adjusted mean and odds ratio.

We have adjusted table 3 to make it clear that we adjusted for calendar time in this analysis.

Table 2 discusses the primary outcome for which adjustment for calendar time is not possible and is consequentially not included in table 2.

VERSION 2 – REVIEW

REVIEWER	Zeng, Irene Auckland University of Technology Faculty of Health and Environmental Sciences, Health Faculty research office
REVIEW RETURNED	12-Jan-2024

GENERAL COMMENTS	Thank you for addressing the comments , particular for your challenges faced during the COVID pandemic. I have some further comments for the revised version, hope to help improving the relevant estimations and methods used. I am happy to help if authors need more time for the clarification. It will be in expectation to add brief justification of using SWR-cluster design, i.e., how to mitigate known bias from SWR-cluster, referencing the protocol. We have adjusted the manuscript to include a justification for the used study design, with reference to the study protocol (p4, lines 122-125):
--

“We have chosen for this design because we hypothesised that the intervention will do more good than harm and we do not want to withhold the intervention from participants until the end of the trial. Additionally, a stepped-wedge design improves the logistical feasibility of the trial and may contribute to successful inclusion. (Groot et al., 2021; Brown, 2016)”

IZ: Thank you for addressing the comments.

2. Elements of SWR-cluster design need to be defined: Clusters and participant, Randomisation and masking, Procedures, Outcomes

In the light of the reviewer’s remarks, we adjusted the manuscript to more clearly define the requested elements based on the CONSORT 2010 guideline extension for stepped wedge cluster randomised trials. We have attached an updated guidance as a supplementary file to our manuscript (#10 CONSORT extension SWCR checklist_complete.pdf)

Clusters are defined as practices (p5, line 134). Participants are defined as patients, GPs and practice assistants of these practices (p5, line 138).

Randomisation took place on the level of the clusters (p10, lines 236-240). We did not perform any masking. The intervention was rolled out sequentially among practices in a stepped-wedge fashion. The primary outcome was measured on patient level and there were no repeated measurements (p9, lines 200-205). Secondary outcomes were measured on the level of practice staff and were measured repeatedly on the individual level (p9, lines 207-211).

In power calculation: authors stated “allowed detection of all standardized effect sizes of 0.12.” how does the trial’s effect size of the subscales in NCQ compare to this in the power calculation? Any potential reason for the smaller effect size? In the design flow chart, it indicates that only GP centers randomized to the first wedge have the intervention for more than 12 months by Dec 2021 -when the second primary outcome were measured. If the duration of the intervention (exposure) affects the effect size, then the GP centers in the second and third wedges have not recieved long enough time to see the effect works because the 12 month outcome were all measured in Dec 2021. For this reason, the duration difference of

GP centers could affect the primary outcome assessment. The duration should be included in the analysis. This could be a potential reason for negative outcome as well, worth including it in the discussion.

3. Statistical analysis: effect of time effect adjustment is not presented in the method.

A. Apart from baseline measures used in the adjustment, any other variables used?

We thank the reviewer for the suggestion to further clarify the used variables for adjustment. The primary outcome was only adjusted for clustering and baseline differences. We have adjusted the manuscript to clarify this. (p11, 257-258):

“The final model was adjusted for clustering and baseline differences and differences in exposure to TOOL-kit between wedges.”

Have the baseline measurement been included as baseline measurement, not baseline difference (what is the baseline difference?). There are three wedges, every GP centre started the intervention at a different calendar month-which should be included in the analysis for **primary outcome** as well, although the results might still be the same, the estimates of the interventional effect will change.

The characteristics of the general practices were likely to affect the outcomes across different wedges, these should be included in the analysis. For example, number of nurses, personal assistants, GPS, very urban/moderate/very rural- even if they were not significantly different between wedges.

B. Temporal effects/time effect (i.e., calendar time) should be included in the adjustment.

We agree with the reviewer that the possible bias due to temporal effects is an important consideration in stepped wedge designs. In line, we have adjusted for temporal effects where possible as is best practice. However, due to the COVID-19 pandemic and its implications for daily general practice it was not possible to perform a sufficient amount of follow-up measurements for the primary outcome to be able to correct for calendar time. We were aware of this limitation and had mentioned this in the discussion

(p18-19, lines 381-389):

“A limitation of our study was the timing and frequency of the NCQ measurements. In stepped-wedge designs, measurements in control conditions are performed earlier than intervention measurements and, as such, calendar time can be a confounder. Initially, we planned to undertake two NCQ follow-up measurements instead of one. However, due to the COVID-19 pandemic and its implications for daily general practice this number of follow-up measurements was regarded as too high of a burden for the participants by both the participating practices and the researchers. Consequently, in the analysis we could not separate the effect of time from the effect of the intervention on personal continuity. However, given the absence of an effect, the attribution question is less relevant and therefore does not influence the overall conclusion of our study.”

For the secondary outcomes, we were able to adjust the analysis for calendar time. We have adjusted the manuscript to make this more clear (p11, lines 268-270):

“The final model was adjusted for clustering, repeated measurements between individuals, baseline differences, differences in exposure to TOOL-kit between wedges, calendar time and baseline differences between wedges.”

Thank you for addressing the comments.

Multiple imputation requires more details, what variables used for imputing the NCQ subscales and why? Please include a reference for the imputation method (i.e., Rubin).

We have adjusted the manuscript to make it more clear which variables were used, why and referenced Rubin’s Rules (p11, lines 261-264):

“We used multiple imputation to handle missing data before calculation the NCQ subscales using all variables in the analysis models. (Heijmans, 2018) These variables were chosen to make optimal use of the observed subscale data. In total, 5 imputed sets were derived. Results were pooled using Rubin’s rules. (Rubin, 1987)”

The method is either monotone or Fully condition specification. The method needs to be specified. Here is the snapshot from SPSS for multiple imputation, where the method could be specified.

Mixed models have not specified distribution of outcome differently for continuous outcome and ordinal outcome. For Example, Linear mixed model for continuous outcome, Generalized linear mixed or generalized estimating equations (GEE) for ordinal outcome.

We have adjusted the manuscript to make this clearer (p11, line 253-254):

“For the primary outcome (continuous), we used linear mixed model analysis to compare the intervention condition (TOOL-kit and usual care) to the control condition (usual care only).”

And (p11, lines 265-266):

“For the secondary outcomes (ordinal), we used a cumulative link mixed model for outcomes on a 5- or 10-point scale to compare the intervention condition to the control condition.”

Thank you for addressing the comments.

If there are same participants included in more than 1 wedge period, random effect should consider include individual participant. The mixed-effect models are highly sensitive to departures from the model assumption, including the random effects. Solutions could be including a time-period random effect.

<https://www.ncbi.nlm.nih.gov/pmc/articles/PMC5600088/>

We thank the reviewer for her critical remark. For the secondary outcome, we have corrected for the repeated measures by adding a random intercept on the individual level (p11, line 266-268). This is in line with the recommendation on page five from the article referenced by the reviewer.

For the primary outcome, this procedure is not applicable as no repeated measures took place. We have adjusted the manuscript to make this clear (p11, lines 256-257):

	“No intercept on the patient level was added as no repeated measures took place on the individual level.” Thank you for addressing the comments. Results: Table 2. and 3 should include temporal effects/time effect (i.e., calendar time) in NCQ subscales, to derive the adjusted mean and odds ratio. We have adjusted table 3 to make it clear that we adjusted for calendar time in this analysis. Table 2 discusses the primary outcome for which adjustment for calendar time is not possible and is consequentially not included in table 2. The results in table 3 is identical comparing to table 3 in the previous version, it does not seem to be revised although the method is now including the calendar time. The odds ratios' 95% confidence intervals are also with typo of “,” instead of “.”. In result description, line 293: “32 general practices were included (median per practice: 6, interquartile range: 4 to 8)”, does the median represent the median number of GP staff of general practices?
--	--

VERSION 2 – AUTHOR RESPONSE

In power calculation: authors stated “allowed detection of all standardized effect sizes of 0.12.” how does the trial’s effect size of the subscales in NCQ compare to this in the power calculation? Any potential reason for the smaller effect size?

The detectable effect size relates to the smallest effect size where our statistical tests would be able to correctly give a statistically significant result. Smaller effect size might go undetected i.e. result in a false negative results. We have reconsidered the meaning of our findings in relation to the clinical significance of the effect size and have added these to the manuscript (20, lines 418-422):

“As the found effect sizes in our study are lower than the effect size in our power calculation, it is possible that the absence of statistical significant findings is a false negative. However, clinical significance should outweigh statistical significance. There is no set minimally clinical significant difference for the NCQ, but the very small effect sizes (≤ 0.05 on a five-point scale) are unlikely to constitute a clinical significant effect.”

In the design flow chart, it indicates that only GP centres randomized to the first wedge have the intervention for more than 12 months by Dec 2021 -when the second primary outcome were measured. If the duration of the intervention (exposure) affects the effect size, then the GP centres in the second and third wedges have not received long enough time to see the effect works because the 12 month outcome were all measured in Dec 2021. For this reason, the duration difference of GP centres could affect the primary outcome assessment. The duration should be included in the analysis. This could be a potential reason for negative outcome as well, worth including it in the discussion

We agree with the reviewer that the difference in exposure to TOOL-kit between wedge may negatively influence the outcome measures. Therefore, we have adjusted the model to reduce this possible bias. This was described in the manuscript (p11, line 258-259):

“The final model was adjusted for clustering and baseline differences and differences in exposure to TOOL-kit between wedges”

A reflection on the limited exposure time of the practices to the intervention was also included in the discussion (p19, lines 390-396):

“It is well known that implementation takes time. Therefore, the follow-up time in our study may have been too short for adequate evaluation of the intervention. This may be especially true for wedges 2 and 3 in which practices only had 15 or 12 months of intervention time respectively, compared with 18 months for wedge 1. However, we found no correlation between intervention time and any outcome measure. Wedge 1 practices did not perform any better on any of the outcome measures than the other wedges. Therefore, it is unlikely that the limited follow-up duration of the last two wedges significantly impacted results.”

Have the baseline measurement been included as baseline measurement, not baseline difference (what is the baseline difference?)

We are not sure we understand the reviewers comment and apologise for any misconceptions. Baseline measurements are often used as covariates instead of outcome measurements. However, this approach is only viable if you follow-up individuals, which we do not. Therefore, we performed only adjustment for the differences in baseline values. We have adjusted the manuscript to make this more clear (p11, lines 257-258):

“baseline measurements were used in the random effects model to adjust for differences in baseline due to clustering”

In addition, we have included the baseline differences in our manuscript (p15, lines 313-316):

“Descriptive statistics shows small baseline differences (‘GP knows me’ mean: 0.02 (SD: 0.17), ‘GP shows commitment’: 0.05 (0.25), team/cross boundary: 0.03 (0.13)) and small

differences between baseline and 12-month follow-up measurements on all three NCQ subscales”

There are three wedges, every GP centre started the intervention at a different calendar month-which should be included in the analysis for primary outcome as well, although the results might still be the same, the estimates of the interventional effect will change.

We agree with the reviewer that calendar month should be included in the analysis. Because each wedge started at a different calendar month (wedge 1; August 2020, wedge 2: November 2020, wedge 3: February 2021), wedge can be seen as a variable associated with calendar month.

We examined the main effects of the wedge and an interaction between wedge and time to investigate heterogeneity in the differences between follow-up/intervention and baseline/control.

We adjusted the manuscript to make this more clear (P11, 260-262):

“Heterogeneity of intervention effect due to longer exposure to TOOL-kit between wedges and temporal effects were investigated using a two-way interaction term.”

The characteristics of the general practices were likely to affect the outcomes across different wedges, these should be included in the analysis. For example, number of nurses, personal assistants, GPS, very urban/moderate/very rural-even if they were not significantly different between wedges.

We agree with the reviewer that the mentioned practice characteristics may influence the outcomes between wedges. Therefore, we have performed additional analyses to screen for confounders. The results are displayed in the table below.

Supplementary table 1: Primary outcome confounder analysis. Relevant covariates were added one by one to the model. Confounding was defined as a >10% shift of the effect from the main model.

Personal continuity: GP knows me					
	Mean difference Follow-up vs baseline	95% CI lower	95% CI upper	P- value	Effect shift
Main model ^a	0,049	-0,052	0,151	0,339	
Practice area (rural v.s. urban)	0,046	-0,055	0,147	0,37	6%
Number of GPs	0,052	-0,049	0,153	0,315	-6%
Number of practice assistants	0,05	-0,52	0,151	0,337	-2%
Number of practice nurses	0,05	-0,052	0,151	0,336	-2%
All covariates	0,049	-0,052	0,151	0,339	0%
Personal continuity: GP shows commitment					
	Mean difference Follow-up vs baseline	95% CI lower	95% CI upper	P- value	Effect shift
Main model ^a	0,032	-0,75	0,139	0,562	
Practice area (rural v.s. urban)	0,027	-0,08	0,134	0,488	16%
Number of GPs	0,033	-0,074	0,14	0,542	-3%
Number of practice assistants	0,031	-0,076	0,138	0,565	3%

Number of practice nurses	0,032	-0,075	0,139	0,56	0%
All covariates	0,029	-0,078	0,136	0,591	9%
Team continuity/transmural continuity					
	Mean difference Follow-up vs baseline	95% CI lower	95% CI upper	P- value	Effect shift
Main model^a	0,011	-0,056	0,079	0,741	
Practice area (rural v.s. urban)	0,011	-0,057	0,078	0,754	0%
Number of GPs	0,012	-0,056	0,08	0,73	-9%
Number of practice assistants	0,011	-0,057	0,079	0,749	0%
Number of practice nurses	0,011	-0,056	0,079	0,746	0%
All covariates	0,013	-0,055	0,08	0,716	-18%

GP= general practitioner

All subscales are measured on a 5 point-Likert scale (1: very low. 5: very high).

^a Adjusted for cluster, exposure to TOOL-kit, baseline differences

We adjusted the manuscript to include the conclusion of the confounder analysis (p15, lines 322-324):

“Practice area and the number of GPs, assistants or nurses per practice were examined as possible confounders. While they did slightly modify the point estimate for the treatment effect (supplementary table S1), these changes did not alter the main conclusion.”

The method is either monotone or Fully condition specification. The method needs to be specified. Here is the snapshot from SPSS for multiple imputation, where the method could be specified.

We thank the reviewer for her suggestion and have adjusted the manuscript (p11, lines 264-265):

“In total, 5 imputed sets were derived using fully conditional specification with 50 iterations.”

The results in table 3 is identical comparing to table 3 in the previous version, it does not seem to be revised although the method is now including the calendar time

We apologise to the reviewer for the confusion caused by our previous communication. It appears we have misunderstood the reviewers comment.

Table 3 has always included adjustment for temporal/time effects. In the first version (V9) this was labelled as ‘length of measurement time’. As we understood the reviewers comment, this phrasing was unclear and we changed ‘length of measurement time’ to ‘calendar time’ based on the reviewers comment.

We hope this additional information explains why **table 3** received no further updates and hope this answers the reviewer’s remarks.

The odds ratios' 95% confidence intervals are also with typo of “,” instead of “.”.

We thank the reviewer and have corrected the error.

In result description, line 293: “32 general practices were included (median per practice: 6, interquartile range: 4 to 8)”, does the median represent the median number of GP staff of general practices

We thank the reviewer for her remark and have adjusted the manuscript (p12, line 298):

“A total of 221 GPs, practice assistants and other practice staff from 32 general practices were included (median practice staff per practice: 6, interquartile range: 4 to 8)”

VERSION 3 – REVIEW

REVIEWER	Zeng, Irene Auckland University of Technology Faculty of Health and Environmental Sciences, Health Faculty research office
REVIEW RETURNED	07-Apr-2024
GENERAL COMMENTS	I congratulate the authors and I endorse the publication. There are remaining minor editing required attentions: 1. Table 1 need to specify that results are all based on imputed data, not completed data. (CONSORT recommend using completed data analysis, imputed results is used for sensitivity Comparison).2. The manuscript needs punctuation checking: the citations are outside of the sentence. For example: In Method, Design and setting “successful inclusion. (46,48) During the trial,” “A detailed study protocol has been published elsewhere. (46)” In Statistical analysis “Baseline measurements were used in the random effects model to 259 adjust for differences in baseline due to clustering The final..” A full stop is expected between clustering and The final.3. In supplementary S1, all decimal points use comma, instead of full stop. For example, p value is “0,339”, mean difference “0,049”, instead of 0.339 and 0.049.